# Genome-Wide Association Study of Fiber Diameter in Alpacas

**DOI:** 10.3390/ani13213316

**Published:** 2023-10-25

**Authors:** Manuel More, Eudosio Veli, Alan Cruz, Juan Pablo Gutiérrez, Gustavo Gutiérrez, F. Abel Ponce de León

**Affiliations:** 1Facultad de Agronomía y Zootecnia, Universidad Nacional de San Antonio Abad del Cusco, Cusco 08006, Peru; mmoremontoya@gmail.com; 2Facultad de Zootecnia, Universidad Nacional Agraria La Molina, Lima 15024, Peru; alancruzc@outlook.com (A.C.); apl@umn.edu (F.A.P.d.L.); 3Centro Experimental La Molina, Dirección de Recursos Genéticos y Biotecnología, Instituto Nacional de Innovación Agraria (INIA), Lima 15024, Peru; eudosio.veli77@gmail.com; 4Estación Científica de Pacomarca, Inca Tops S.A., Arequipa 04007, Peru; 5Departamento de Producción Animal, Facultad de Veterinaria, Universidad Complutense de Madrid, 28040 Madrid, Spain; gutgar@vet.ucm.es; 6Instituto de Investigación de Bioquímica y Biología Molecular, Universidad Nacional Agraria La Molina, Lima 15024, Peru; 7Department of Animal Science, University of Minnesota, Minneapolis, MN 55108, USA

**Keywords:** GWAS, SNP, genotyping, fiber diameter, alpaca

## Abstract

**Simple Summary:**

The aim of this study was the identification of candidate genomic regions associated with fiber diameter in alpacas. DNA samples were collected from 1011 female Huacaya alpacas from two geographical Andean regions in Peru (Pasco and Puno), and three alpaca farms within each region. The samples were genotyped using an Affymetrix Custom Alpaca genotyping array containing 76,508 SNPs. After the quality controls, 960 samples and 51,742 SNPs were retained. Three association study methodologies were performed. Four candidate regions with adjacent SNPs identified via two association methods of analysis are located on VPA6, VPA9, VPA29 and one chromosomally unassigned scaffold. This study represents the first analysis of alpaca whole genome association with fiber diameter, using a recently developed alpaca SNP microarray.

**Abstract:**

The aim of this study was the identification of candidate genomic regions associated with fiber diameter in alpacas. DNA samples were collected from 1011 female Huacaya alpacas from two geographical Andean regions in Peru (Pasco and Puno), and three alpaca farms within each region. The samples were genotyped using an Affymetrix Custom Alpaca genotyping array containing 76,508 SNPs. After the quality controls, 960 samples and 51,742 SNPs were retained. Three association study methodologies were performed. The GWAS based on a linear model allowed us to identify 11 and 35 SNPs (−log_10_(*p*-values) > 4) using information on all alpacas and alpacas with extreme values of fiber diameter, respectively. The haplotype and marker analysis method allowed us to identify nine haplotypes with standardized haplotype heritability higher than six standard deviations. The selection signatures based on cross-population extended haplotype homozygosity (XP-EHH) allowed us to identify 180 SNPs with XP-EHH values greater than |3|. Four candidate regions with adjacent SNPs identified via two association methods of analysis are located on VPA6, VPA9, VPA29 and one chromosomally unassigned scaffold. This study represents the first analysis of alpaca whole genome association with fiber diameter, using a recently assembled alpaca SNP microarray.

## 1. Introduction

Alpaca fiber is the main source of income for alpaca producers. It is in high demand by the textile industry, and is a valuable export product for Peru. Alpaca breeding programs implemented by private farms and farm community cooperatives have the goal of improving fiber quality by decreasing fiber diameter. Genetic gain for fiber production and fiber quality traits could increase rapidly with the implementation of marker-assisted selection. However, an efficient application of this genomic tool requires the identification of molecular markers and genomic regions associated with fiber traits.

Two research groups have sequenced the alpaca genome, generating depths of 72.5X [1] and ∼190X [2]. The alpaca reference genome has been assembled into 77,390 scaffolds (VicPac3.1, GCA_000164845.4, NCBI), of which 88 scaffolds are assigned to the 36 autosomal chromosomes and the X chromosome, representing 76% of the reference genome length [2].

Chromosomal syntenies among bovines, humans and camelids using Zoo-FISH were reported by Balmus et al. [3]. The first cytogenetic map for molecular markers and genes was developed using fluorescence in situ hybridization [4] and expanded [5,6,7]. Using a high-density bovine single nucleotide polymorphism (SNP) microarray to genotype alpacas, More et al. [8] uncovered 6756 conserved SNPs between bovines and alpacas. Other studies have reported SNPs and other variants identified at candidate genes for fiber quality and color traits [9,10,11,12,13,14]. A recent study reported the development of an alpaca 76K SNP microarray [15], which is now available.

The application of genome-wide association studies (GWASs) to identify genomic regions associated with fiber and/or wool traits and the discovery of regulatory and/or structural variants in genes involved in hair follicle development have been reported for different species. In Merino sheep, Wang et al. [16] identified nine SNPs significantly associated with fiber diameter. In Alpine Merino, Chinese Merino, Qinghai and Aohan sheep, Zhao et al. [17,18] used sequencing data to identify SNPs with a significant association with fiber diameter at 1 year of age, reporting two SNPs in a first study and seven SNPs in a second study with a larger population. In Merino sheep, Bolormaa et al. [19] identified 43 independent SNPs significantly associated with fiber diameter at 550 days of age or older, using a genome-wide association study with multiple traits. In cashmere goats, Qiao et al. [20] did not identify SNPs significantly associated with fiber diameter, but they identified 26 SNPs with the highest −log_10_ *p*-values, with some located near genes involved in signaling pathways associated with development of hair follicles and genes related to hair follicles and skin. In another association study, Wang et al. [21] identified eight SNPs associated with fiber diameter in Inner Mongolia cashmere goats. Li et al. [22] identified 135 genomic regions under selection in cashmere goats, including genes associated with fiber characteristics, based on the values of the fixation index (F_st_) and nucleotide diversity (θπ). In addition, a recent study identified selection signals when comparing the F_st_ and cross-population extended haplotype homozygosity (XP-EHH) values of a non-cashmere goat breed and two cashmere goat breeds [23], including genes related to 67 signaling pathways involved in the development of hair follicles. In an elite population of Huacaya and Suri alpacas from Peru, a region-specific association study identified four microsatellite loci (*LCA68*, *VOLP59*, *LCA90* and *GLM6*) significantly associated with genetic merit for fiber diameter [24]. However, genome-wide association studies (GWASs) have not been applied to alpacas due to the limited number of available SNP genotyping platforms.

Based on the experiences with other species and the availability of an alpaca 76K SNP microarray, this study aimed to apply GWASs to identify candidate genomic regions associated with fiber diameter in alpacas.

## 2. Materials and Methods

### 2.1. Ethics Statement

Alpaca blood samples for DNA isolation were obtained in accordance with the Peruvian National Law No. 30407, “Animal Protection and Welfare Law”, in effect in Peru since 7 January 2016. The Dean of the College of Animal Science approved the protocol in lieu of the UNALM “Ethics Committee for Scientific Research” No. 0345-2018-CU-UNALM—22 October 2018, which has not started operations.

### 2.2. DNA Samples and Genotyping

A sample of 1011 female Huacaya alpacas originating from two geographical Andean regions, Puno in the south and Cerro de Pasco in the central Andes, and three alpaca farms within the regions, was identified. All the alpacas had a white coat phenotype. Fiber diameter records at second shearing were available for these animals. The fiber diameter was evaluated using the Sirolan Laserscan according to International Wool Textile Organization standards (IWTO-12) [25].

Blood samples (6 to 10 mL) were obtained via venipuncture and maintained in vacutainer tubes containing 10.8 mg of EDTA-K2. DNA extraction using the PureLink^TM^ Genomic DNA Mini kit (Invitrogen, Carlsbad, CA, USA) was performed at Instituto Nacional de Innovación Agraria (INIA). Samples were genotyped using an Affymetrix Custom Alpaca genotyping array containing 76,508 SNPs [15]. Genotyping was performed at NEOGEN GeneSeek laboratories (Nebraska, Lincoln, NE, USA).

### 2.3. Data Quality Control

The first quality control was performed using Axiom Analysis Suite v.4.0.3.3 software (Affymetrix, Santa Clara, CA, USA). The quality control by sample included Dish QC (DQC) ≥ 0.82. The quality control by SNP included the best and recommended SNPs (PolyHighResolution, NoMinorHom, MonoHighResolution).

For the association analysis, a second quality control was performed using PLINK v1.90p software [26]. All SNPs with a genotyping rate ≥90%, a minor allelic frequency (MAF) ≥ 0.05 and a Hardy-Weinberg equilibrium (HWE) *p*-value > 1 × 10^−6^ were retained.

### 2.4. Genome-Wide Association Study

The association studies included three methodologies: (1) a GWAS based on a linear model [27], (2) haplotype and marker analysis [28] and (3) selection signatures based on cross-population extended haplotype homozygosity (XP-EHH) [29].

#### 2.4.1. Method 1. Linear Model

The significant fixed effects on fiber diameter at second shearing were identified using analysis of variance. According to these results and before performing the association analysis, the fiber diameter at second shearing was corrected using herd and month-year of shearing as fixed effects and age as a linear covariate and considering all the data from the sample of 960 alpacas retained after quality control of genotyping data, using SAS on Demand for Academics software (SAS Institute Inc., Cary, NC, USA).

A first approach was developed using all the alpacas. The SNP quality control was performed as previously described in the section “Data Quality Control”. A total of 51,742 SNPs in 960 female Huacaya alpacas were retained.

The association analysis based on a mixed linear model was performed considering the genomic relationships to correct for the polygenic effect of the SNPs. The genomic relationship matrix was generated from the genotyping data on SNPs using the GCTA software’s [30] “make-grm” option. The model equation [27] was
(1)$$y=X_{\mathrm{snp}}\beta_{\mathrm{snp}}+g+e,$$
where **y** is the corrected fiber diameter at second shearing vector, **X**_snp_ is the genotype matrix, **β**_snp_ is the effect of each SNP, **g** is a vector of the total genetic effects based on the matrix of genomic relationships and **e** is the vector of residuals.

A second approach was developed using alpacas from two extreme groups in terms of fiber diameter at second shearing. A group of 112 alpacas with fiber diameter less than or equal to 18 microns (fine group) and another group of 115 alpacas with fiber diameter greater than 23 microns (coarse group) were identified. Each pair of alpacas with genomic relationships greater than 0.10 was identified, and only one alpaca of each pair was retained [31] using the GCTA software’s [30] “grm-singleton” option. A total of 104 unrelated alpacas were retained, of which 49 belonged to the fine group and 55 belonged to the coarse group. The quality control was performed using PLINK v1.90p software [26]. All SNPs with a genotyping rate lower than 0.90, a minor allelic frequency (MAF) lower than 0.05 and an HWE *p*-value lower than 1 × 10^−6^ were removed. A total of 59,949 SNPs in 104 female Huacaya alpacas were retained.

The association analysis based on a linear model was performed without considering the genomic relationships because only unrelated alpacas were included. The model equation was
(2)$$y=X_{\mathrm{snp}}\beta_{\mathrm{snp}}+e,$$
where **y** is the corrected fiber diameter at second shearing vector, **X**_snp_ is the genotype matrix, **β**_snp_ is the effect of each SNP and **e** is the vector of residuals.

Both analyses were performed using the fastGWA method with GCTA software [27]. The significance threshold for the GWASs was defined using the Bonferroni correction method [21], considering an error rate of 0.05 and the number of analyzed SNPs. The SNPs with −log_10_(*p*-values) lower than the significance threshold (6.01) were considered insignificant SNPs. Moreover, the SNPs with (−log_10_(*p*-values) > 4) were selected as candidate SNPs. The genes harboring selected SNPs as well as the genes found within a span of 500 kbp 5′ and 3′ flanking each of the SNPs were identified using the annotation of the alpaca reference genome VicPac3.1.

#### 2.4.2. Method 2. Haplotype and Marker Analysis

A total of 51,742 SNPs in 960 female Huacaya alpacas were retained after the quality controls. Haplotype phasing and imputation of missing genotypes were performed before the haplotype and marker analysis. Because a genetic map for alpacas is not available, the imputation by scaffold was performed assuming that 1 centiMorgan is equal to 1 Mbp. The imputation was performed using Beagle 5.1 v25Nov19 software [32,33].

The haplotype blocks were defined considering that two SNPs are under “strong” linkage disequilibrium if the lower level of the confidence interval (90% confidence value) of the value of D’, defined as the ratio between the linkage disequilibrium coefficient (D) and the maximum value of D for the given allele frequencies (Dmax), is greater than 0.70, and the upper level of the confidence interval is at least 0.98 [34,35]. The blocks were generated using PLINK v1.90p software [26], option “blocks”.

The model included the additive effect of the SNP, the dominance effect and the additive effect of the haplotype. The model equation [28] was
(3)$$y=Xb+Z\left(W_{\alpha }\alpha_{o}+W_{\delta }\delta_{o}+W_{\alpha h}\alpha_{h}\right)+e,$$
where **y** is the fiber diameter at second shearing vector, **Z** is an incidence matrix that relates the phenotypic records with each animal, **α**_o_ is a column vector for additive genetic effects of the SNPs, **W**_α_ is an incidence matrix for **α**_o_, **δ**_o_ is a column vector for dominance effects, **W**_δ_ is an incidence matrix for **δ**_o_, **α**_h_ is a column vector for additive haplotype genetic effects, **W**_αh_ is an incidence matrix for **α**_h_, **b** is a column vector for herd and month–year fixed effects of shearing and age as a linear covariate and **X** is an incidence matrix for b. The estimation of variance components and heritabilities was performed using GVCHAP software [28]. The haplotypes with the highest standardized heritability (>6) were selected. The genes within these haplotypes as well as the genes found within a span of 500 kbp of the haplotypes were identified using the annotation of the alpaca reference genome VicPac3.1.

#### 2.4.3. Method 3. Selection Signatures

A total of 104 unrelated alpacas from two extreme groups in terms of fiber diameter at second shearing were identified as previously described in the section “Method 1. Linear Model”. The fiber diameters at second shearing of the fine group and coarse group were 17.16 ± 0.76 and 24.38 ± 1.39 microns, respectively. The difference in the means between the two groups was significant (*p*-value < 0.0001).

From the imputed data in method 2, the genotyping data on the 104 alpacas were retrieved. The quality control was performed using PLINK v1.90p software [26]. All SNPs with a genotyping rate lower than 1.0, a minor allelic frequency (MAF) lower than 0.05 and an HWE *p*-value lower than 1 × 10^−6^ were removed. A total of 49,306 SNPs in 104 female Huacaya alpacas were retained. The reference (higher frequency) and alternative (lower frequency) alleles were defined based on the original population of 960 female Huacaya alpacas.

The identification of selection signatures based on cross-population extended haplotype homozygosity (XP-EHH), where population A is the coarse group and population B is the fine group (reference population), was performed. The statistics were calculated using Selscan v1.3.0 software [36]. The analysis was performed within each scaffold, considering a haplotype length of up to 1 Mbp and the presence of a gap of up to 200 kbp between SNPs. The values of XP-EHH for each locus were standardized across all the scaffolds by subtracting the mean and dividing the result by the standard deviation.

The selection signatures with standardized values of XP-EHH > |3| were selected. The genes harboring these SNPs as well as the genes found within a span of 500 kbp of the SNPs were identified using the annotation of the alpaca reference genome VicPac3.1.

### 2.5. Candidate Regions

Candidate regions were identified based on adjacent SNPs at a maximum intermarker distance of 500 kbp from one or more methodologies. The genes annotated within these regions as well as the genes annotated within a span of 500 kbp 5′ and 3′ of these regions were identified using the annotation of the alpaca reference genome VicPac3.1. The gene ontology (GO) annotations (hair follicle morphogenesis, hair follicle development, skin morphogenesis, skin development and keratin filament) of *Vicugna pacos*, *Ovis aries*, *Capra hircus*, *Mus musculus* and *Homo sapiens* were retrieved from QuickGO [37], and the Kyoto Encyclopedia of Genes and Genomes (KEGG) pathways of *Ovis aries* were retrieved from the Database for Annotation, Visualization, and Integrated Discovery (DAVID, [38]). A functional annotation clustering was performed using DAVID Bioinformatics Resources 2021, considering an *Ovis aries* background because a complete *Vicugna pacos* background is not available on the DAVID database. The clusters with enrichment scores higher than or equal to 1.3 were selected [39,40].

## 3. Results

### 3.1. Data Quality Control

After the first quality control, 960 samples and 58,865 SNPs were retained. After the second quality control, a total of 51,742 SNPs, located in 1432 scaffolds, were retained. The length of these scaffolds represents 95.34% of the total length of the alpaca genome. According to their location, 611 SNPs are located in 7 scaffolds assigned to the X chromosome, 41,048 SNPs in 78 scaffolds assigned to the 36 autosomal chromosomes and 10,083 SNPs in 1347 scaffolds not assigned to chromosomes. The lengths of these scaffolds represent 1.62%, 73.98% and 19.74% of the total length of the alpaca genome, respectively.

The fiber diameters at second shearing of the 960 samples were 20.53 ± 2.17 microns, and ranged from 13.73 to 29.50 microns.

### 3.2. Genome-Wide Association Study

#### 3.2.1. Method 1. Linear Model

A set of 11 SNPs (−log_10_(*p*-values) > 4) was selected using a sample of 960 alpacas (Figure 1 and Appendix A). The SNP effects with values preceded by a negative sign (Appendix A) indicate that the presence of the less frequent allele could be associated with a decrease in fiber diameter. A total of 18 lncRNA and 69 protein-coding genes harbored these SNPs or were found within a span of 500 kbp from the identified SNPs (Appendix A). The GO annotations were not identified. We present the KEGG pathways in Appendix A. The KEGG pathways were not associated significantly with the genes.

A region of 290.9 kbp with five identified SNPs was located at a chromosomally unassigned scaffold (Appendix A). A total of three lncRNA and five protein-coding genes harbored these SNPs or were found within a span of 500 kbp from the identified SNPs (Appendix A).

A set of 35 SNPs (–log_10_(*p*-values) > 4) was selected using the sample representing extreme groups in terms of fiber diameter (Figure 2 and Appendix A). The SNP effects with values preceded by a negative sign (Appendix A) indicate that the presence of the less frequent allele could be associated with a decrease in fiber diameter. A total of 73 lncRNA and 235 protein-coding genes harbored these SNPs or were found within a span of 500 kbp from the identified SNPs (Appendix A). We present the GO annotations in Appendix A. The GO annotations were not associated significantly with the genes. We present the KEGG pathways in Appendix A. The KEGG pathways were not associated significantly with the genes.

A region of 315.9 kbp flanked by three identified SNPs was located on VPA4. A total of two lncRNA and two protein-coding genes harbored these SNPs or were found within a span of 500 kbp from the identified SNPs. Another region of 121.2 kbp flanked by two identified SNPs was located on VPA14. A total of three lncRNA and nine protein-coding genes harbored these SNPs or were found within a span of 500 kbp from the identified SNPs. Another region of 17.3 kbp flanked by two identified SNPs was located on VPA18. A total of 1 lncRNA and 22 protein-coding genes harbored these SNPs or were found within a span of 500 kbp from the identified SNPs. Another region of 53.4 kbp flanked by three identified SNPs was located on VPA32. A total of 7 lncRNA and 26 protein-coding genes harbored these SNPs or were found within a span of 500 kbp from the identified SNPs. Moreover, another region of 157.8 kbp flanked with two identified SNPs was located at a chromosomally unassigned scaffold. A total of three lncRNA and four protein-coding genes harbored these SNPs or were found within a span of 500 kbp from the identified SNPs. Of these, one protein-coding gene (*PRR16*) was common with the analysis of all the alpacas.

#### 3.2.2. Method 2. Haplotype and Marker Analysis

Of the 3160 scaffolds with SNPs included in the alpaca microarray [15], 1432 scaffolds had SNPs that passed the second quality control. Of these scaffolds, 1211 scaffolds were excluded because the imputation requires at least two SNPs per scaffold. These excluded scaffolds represent approximately 0.43% of the alpaca Vicpac3.1 reference genome. The imputation was performed on the remaining 221 scaffolds containing 50,531 SNPs. The length of these scaffolds represents 94.91% of the total length of the alpaca genome.

All SNPs located on 28 scaffolds and some SNPs on the remaining 193 scaffolds were not considered within the haplotype blocks because these SNPs were not under “strong” linkage disequilibrium. Hence, a total of 6244 haplotype blocks (Appendix A) including 20,438 SNPs located on 193 scaffolds were defined as previously described in the section “Method 2. Haplotype and Marker Analysis”. The haplotype blocks contained, on average, 3.27 ± 2.47 SNPs. The analysis included the effect of haplotypes (6244) and the individual effect of SNPs (50,531).

A set of nine haplotypes containing 27 SNPs was identified with standardized haplotype heritability higher than six standard deviations (Figure 3, Appendix A). These haplotypes were localized on chromosomes VPA2 (1), VPA7 (1), VPA11 (1), VPA14 (1) and VPA24 (2) and three chromosomally unassigned scaffolds (3). A total of 15 lncRNA and 68 protein-coding genes harbored these haplotypes or were found within a span of 500 kbp from the identified haplotypes (Appendix A). The GO annotations were not identified. We present the KEGG pathways in Appendix A. The KEGG pathways were not associated significantly with the genes.

A haplotype of 12.7 kbp located on VPA24 and two haplotypes spanning 72.2 kbp and 110.2 kbp, respectively, located on two chromosomally unassigned scaffolds had the highest standardized heritability (>8.0 standard deviations, SD).

#### 3.2.3. Method 3. Selection Signatures

Standardized XP-EHH values greater than 2 SD or lower than −2 SD were used to identify 1954 selection signals (Figure 4). The positive scores indicate selection in the observed population (coarse group), and the negative scores indicate selection in the reference population (fine group) [41].

A set of 95 SNPs was identified with standardized XP-EHH values >3 (Appendix A). The regions with the highest numbers of SNPs with standardized XP-EHH values >3 were located at VPA6 (15 SNPs), VPA4 (18 SNPs) and VPA5 (11 SNPs). A total of 70 lncRNA and 315 protein-coding genes harbored these SNPs or were found within a span of 500 kbp from the identified SNPs (Appendix A). We present the GO annotations in Appendix A. The GO annotations were not associated significantly with the genes. We present the KEGG pathways in Appendix A. The KEGG pathways associated significantly (*p*-value < 0.05) with the genes were glycosaminoglycan biosynthesis—keratan sulfate, thyroid hormone synthesis and spliceosome.

A set of 85 SNPs was identified with standardized XP-EHH values <−3 (Appendix A). The regions with the highest numbers of SNPs with standardized XP-EHH values <−3 were located at VPA12 (29 SNPs) and VPA4 (15 SNPs). A total of 81 lncRNA and 267 protein-coding genes harbored these SNPs or were found within a span of 500 kbp from the identified SNPs (Appendix A). We present the GO annotations in Appendix A. The GO annotations were not associated significantly with the genes. We present the KEGG pathways in Appendix A. The KEGG pathways associated significantly (*p*-value < 0.05) with the genes were N-glycan biosynthesis, metabolic pathways, histidine metabolism and nucleocytoplasmic transport.

We reported the results with standardized XP-EHH values > |3|. However, in regions with SNPs identified using a lower level of significance, we found keratin genes associated with hair. Type II hair keratin genes (*KRT82*, *KRT83*, *KRT84*) and a keratin gene specific for the inner root sheath of the hair follicle (*KRT72*) were found in regions with SNPs identified using standardized XP-EHH values > 2. A type I hair keratin gene (*KRT35*) and a keratin gene specific for the inner root sheath of the hair follicle (*KRT26*) were found in regions with SNPs identified using standardized XP-EHH values <−2.

### 3.3. Candidate Regions

A set of 251 SNPs was identified using at least one methodology, and these SNPs were located in 92 regions. A total of 46 candidate regions formed by two or more adjacent and identified SNPs were defined (Appendix A), considering a maximum intermarker distance of 500 kbp. A total of 138 lncRNA and 498 protein-coding genes harbored these regions or were found within a span of 500 kbp from the regions (Appendix A). The GO annotations were not associated significantly with the genes. We present the KEGG pathways in Appendix A. The KEGG pathways associated significantly (*p*-value < 0.05) with the genes were metabolic pathways and N-glycan biosynthesis.

Four regions were flanked by SNPs identified using two methods (Table 1). A total of 13 lncRNA and 40 protein-coding genes harbored these regions or were found within a span of 500 kbp from the regions (Appendix A). The GO annotations were not associated significantly with the genes. We present the KEGG pathways in Appendix A. The KEGG pathways were not associated significantly with the genes.

## 4. Discussion

### 4.1. Sample

The ideal sample to develop GWASs in alpacas should be made up of unrelated animals with their respective fiber diameter records at second shearing. The identification of animals with these characteristics is limited in alpacas in comparison with other species due to the restricted use of pedigree and production records, and the limited use of reproductive biotechnologies such as artificial insemination. At least two private farms have production records for four to six generations. And a handful of private and community farms have production records for one or two generations. However, most alpaca farms do not maintain pedigree and production records, and a significantly smaller number of alpaca farms keep records for fiber characteristics and/or perform fiber diameter analysis. Hence, our sample was affected by these realities.

### 4.2. Genome-Wide Association Study

Population stratification [42,43,44] and animal relatedness [45,46] are considered confounding factors in association studies, likely leading to spurious associations and false positives. Several methodologies have been essayed here with different assumptions, which is useful in dealing with these issues.

Minimizing spurious associations can be achieved, for example, by including the genomic relationship matrix in the model jointly with the genotype to be checked. The inclusion of the genomic relationship matrix in a mixed linear model controls the inflation of association test statistics due to relatedness, and it is not necessary to exclude related individuals [27,44]. The heat map of genomic relationships (Figure 5) shows on the diagonal line six subpopulation blocks, Gacocen, Mallkini, Pacomarca, Quimsachata, Racco and Sogamu, where it is clearly visualized that Gacocen, Racco and Sogamu showed more animals with common ancestors within a farm. Gacocen and Racco showed more animals with common ancestors between farms. Mallkini and Pacomarca also showed ancestry that is more common within a farm but not between farms. The proportion of genomic relationships >0.1 is 11.19% for Gacocen, 2.52% for Mallkini, 4.51% for Pacomarca, 5.71% for Quimsachata, 14.68% for Racco and 22.96% for Sogamu.

For the linear model analysis using extreme groups and the selection signature analysis based on XP-EHH, we excluded related individuals to control for relatedness. We calculated the relationships between individuals based on genomic information, and filtered out one of each pair of closely related individuals [31]. Other reports defined a close relationship based on a threshold of 0.2 [47]. We instead considered a higher stringent threshold of 0.1.

### 4.3. Discussion by Method

#### 4.3.1. Method 1. Linear Model

Öner et al. [48] reported the development of GWASs in Assaf sheep using animals with extreme values (the highest 10% and the lowest 14% values) of somatic cell count, and the validation of the results using all the analyzed animals. Li et al. [49] reported the development of GWASs in beef cattle using all the animals and GWASs using animals with extreme values (different combinations of high and low percentiles) of yearling weight. We developed GWASs using all the alpacas, and GWASs using alpacas with extreme values (the lowest 12% and the highest 12%) of fiber diameter at second shearing. SNPs identified with both linear model analyses were not significant because their *p*-values were higher than the threshold (9.6 × 10^−7^) calculated based on the Bonferroni test. However, we selected the SNPs with *p*-values less than 1 × 10^−4^ [50]. Moreover, SNPs located at VPA6, VPA9, VPA29 and one unassigned scaffold are close to the SNPs identified using other methodologies (Table 1). One SNP identified using all the alpacas and located at VPA16 had the greatest effect on fiber diameter at second shearing, with a standardized SNP effect of 5.99 standard deviations (SD). Another SNP identified using extreme alpacas and located at VPA18 had the greatest effect on fiber diameter at second shearing, with a standardized SNP effect of 6.70 SD.

In a complementary analysis, a set of three significant SNPs (*p*-values less than 9.98 × 10^−7^) located at VPA1, VPA19 and an unassigned scaffold was identified using extreme groups from Pacomarca. This analysis was not included in the Materials and Methods section because only 51 unrelated alpacas (small size sample) were retained, of which 20 belonged to the fine group and 31 belonged to the coarse group.

Our results suggest that many SNPs with small effects influence fiber diameter at second shearing. However, a new analysis with a larger number of samples would be required to confirm the presence of SNPs with larger effects. The availability of animals with complete records is limited in alpacas because alpaca producers do not maintain pedigree records and/or production records nor perform fiber diameter analysis. Future studies require generating phenotypic and genomic data from new alpaca generations. On the other hand, fiber diameter analysis was performed in three different laboratories, which in itself generates a confounding effect that could not be controlled in the present study.

#### 4.3.2. Method 2. Haplotype and Marker Analysis

The grouping of SNPs to build haplotype blocks has been based on fixed distance, linkage disequilibrium [35], a fixed number of SNPs [51] or a combination of the number of SNPs and linkage disequilibrium [52]. We built haplotype blocks based on strong linkage disequilibrium [34], which reduces the number of explanatory variables used in the analysis [53]. This methodology generated a variable number of SNPs by haplotype block when compared with other methodologies that are based on a fixed number of SNPs. A disadvantage of the method we used is the significant reduction in genome coverage of the haplotypes due to the exclusion of regions with low linkage disequilibrium. However, our analysis included the individual effects of 20,438 SNPs found within haplotype blocks and 30,093 SNPs found outside of the haplotype blocks.

The haplotype and marker analysis allowed us to identify regions based on standardized haplotype heritability >6. Of 27 SNPs located in the identified haplotypes, none had SNP heritability >6; therefore, these regions may not be identified when evaluating only SNP heritability [28].

#### 4.3.3. Method 3. Selection Signatures

The use of extreme values of XP-EHH has been reported for the identification of selection signatures related to wool or fiber traits and adaptation traits in sheep and goats. Jin et al. [23] identified SNPs with the top 5% of the XP-EHH values using SNP microarray genotyping data on 53 goats from three breeds. Lei et al. [54] identified SNPs with XP-EHH <−2 using available SNP microarray genotyping data on 795 sheep from 27 breeds divided into two groups (fine-wool sheep and hair sheep). Zhang et al. [55] identified SNPs with the top 1% of the XP-EHH values using sequencing data on 47 sheep from three breeds. Abied et al. [56] identified SNPs with the top 1% of the XP-EHH values using high-density SNP microarray genotyping data on 96 sheep from five Chinese sheep breeds. In our study, the SNPs under selection were identified based on XP-EHH higher than |3|, equivalent to SNPs with the top 1% of the XP-EHH values. This stringent value was used to identify the SNPs with the most extreme values, significantly reducing the number of signals compared with the threshold of |2|. This methodology allowed us to contrast the fine fiber and coarse fiber groups and identify selection signatures in alpacas.

### 4.4. Gene Annotation

Song et al. [57] identified two PCR-SSCP patterns located in *KRTAP1-3* associated with a higher mean fiber diameter in Longdong cashmere goats (*p*-value < 0.001 and *p*-value < 0.01). This goat gene is orthologous to *LOC102529268* (keratin-associated protein 4-3-like) located on VPA16 in alpacas. This gene was found within a span of 500 kbp 5′ flanking six SNPs identified with standardized XP-EHH <−2 located at position NW_021964192.1:622090-636205.

The GO annotations for hair follicle morphogenesis and hair follicle development (Appendix A) and the KEGG pathways MAPK, Ras, Notch, PI3K-Akt, Jak-STAT, Wnt and TNF signaling pathway (Appendix A) of the genes in the regions with identified SNPs were not significant. Association studies in cashmere goats [20,21,23] reported genes related to MAPK, Ras, Notch, PI3K-Akt, Wnt and TNF signaling pathways, associated with hair follicles. Zhao et al. [58] reported that PI3K-Akt, Jak-STAT and MAPK signaling pathways may participate in the regulation of wool growth in sheep. Lv et al. [59] reported target genes of lncRNA that were enriched in keratin filament, MAPK, PI3K-Akt and Ras signaling pathways, using an analysis of expression in hair follicles of sheep. Rishikaysh et al. [60] reported that the morphogenesis of hair follicles depends on different signaling pathways, such as Wnt, Shh, Notch and BMP. Moreover, Chen et al. [61] reported that the PI3K-Akt signaling pathway is involved in de novo hair follicle regeneration.

The KEGG pathways associated significantly with the genes were metabolic pathways and N-glycan biosynthesis. Zhao et al. [62] reported an association between candidate genes for wool traits in sheep, such as fiber diameter, and metabolic traits in humans. During hair follicle development in sheep, Guo et al. [63] reported genes enriched in the glycolysis/gluconeogenesis pathway.

The significance value for GO annotation and KEGG pathway analysis was calculated using DAVID [38] and the *Ovis aries* database. Because of the use of a database from another species with better annotation in comparison with alpacas, some alpaca genes were not found in the sheep database, hence these genes were not included in the analysis.

### 4.5. Candidate Regions

Association studies in sheep identified significant markers for fiber diameter (Appendix A) in regions within or close to genes located on OAR1, OAR2, OAR14, OAR17 and OAR25 [16,17,18,19]. These genes are annotated on VPA1, VPA5, VPA9, VPA2/VPA32 and VPA11 in the alpaca. Comparative Zoo-FISH studies between bovine and sheep genomes [64,65] and studies between camel and bovine genomes [3] provide support for synteny between these sheep and alpaca chromosomes. We identified SNPs in these chromosomes, but they were located at other loci. The latter could be due to intrachromosomal rearrangements which are not identifiable with Zoo-FISH, and therefore the chromosomal syntenies, while valid for whole chromosomes and large chromosomal segments, might not be valid for small chromosomal segments.

An association study in goats identified significant markers for fiber diameter (Appendix A) in regions close to genes located on CHI4, CHI21 and CHI28 [20]. These genes are annotated on VPA7, VPA27/VPA6 and VPA11 in the alpaca. Another association study in goats identified significant markers for fiber traits (Appendix A) in regions within or close to genes located on CHI4, CHI6 and CHI15 [21]. These genes are annotated on VPA7, VPA2 and VPA10 in the alpaca. A selection signature study in goats identified significant markers for fiber traits (Appendix A) in regions within or close to genes located on CHI6 and CHI14 [22]. These genes are annotated on VPA2 and VPA29 in the alpaca. Moreover, a selection signature analysis identified significant regions (Appendix A) within or close to genes located on CHI2, CHI6, CHI10, CHI25 and CHI29 [23]. These genes are annotated on VPA5, VPA2, VPA6, VPA18 and VPA10 in the alpaca. Comparative Zoo-FISH studies between bovine and goat genomes [65,66], studies between human and goat genomes [66] and studies between camel and bovine genomes [3] once again provide support for syntenies among these interspecies chromosome sets. However, we must emphasize that the development of Zoo-FISH experiments is necessary to validate these synteny blocks. In our study, most of the SNPs were identified in different regions within these chromosomes; however, one SNP was located in regions close (<200 kbp) to the gene *NTRK3* (VPA27) and three markers were located in regions close (<500 kbp) to the genes *TACC3* (VPA2), *PSMA2* and *POLD2* (VPA7). The information about these regions is reported in Table 2.

### 4.6. Genome Annotation

On the reference genome Vicpac3.1, 18 chromosomes are fragmented in 2 or 3 scaffolds, the chromosomes 1, 11, 16, 35 and X are fragmented in 4 or more scaffolds [2], and there are more than 77,000 chromosomally unlocalized small scaffolds. Moreover, Richardson et al. [2] reported 42,389 predicted genes in the alpaca genome, of which 22,462 were coding genes and 19,927 predicted genes did not have similarity with peptides on the mammalian RefSeq database [67]. The genes harboring the SNPs identified in this study as well as the genes found within a span of 500 kbp of the SNPs were identified using the annotation of the alpaca reference genome VicPac3.1; however, some SNPs could be localized at these unreported predicted genes. Updates to the assembly and annotation are needed to confirm the existence of these genes in alpacas.

### 4.7. Future Studies

The validation of the identified genetic markers should be the next step. Subsequent GWASs should incorporate a higher number of alpacas to confirm the association of SNPs with fiber diameter. Moreover, since the alpaca herds were poorly genetically connected [15], both the allelic frequency of SNPs and the association with fiber diameter in other populations must be evaluated, in order to know whether the markers are valid for any population or only for the populations studied. After the validation of the genetic markers, genotyping tests must be developed to apply genomic selection for reducing fiber diameter.

## 5. Conclusions

This is the first alpaca whole genome association study of fiber diameter using a recently assembled SNP microarray. We identified candidate genome regions associated with fiber diameter in alpacas. Four candidate regions with adjacent SNPs identified using two association methods of analysis are located on VPA6, VPA9, VPA29 and one chromosomally unassigned scaffold. SNPs located on VPA2, VPA7 and VPA27 and identified using one association method of analysis were localized within or next to orthologous alpaca genes that had been reported as candidate fiber genes in goats, and could be considered as candidate fiber diameter-associated SNPs in alpacas. The alpaca SNP array successfully identified alpaca SNPs and chromosomal regions associated with fiber diameter. These results represent the first contribution to the implementation of genetic improvement programs based on marker-assisted selection in alpacas. In the future, other association studies of characteristics such as fleece weight and medullation percentage could be performed.

## Figures and Tables

**Figure 1 animals-13-03316-f001:**
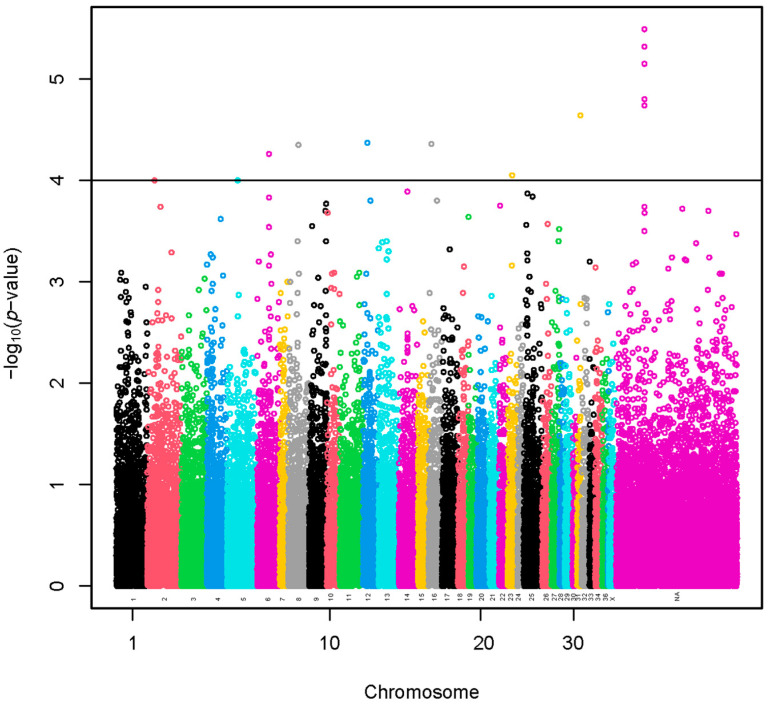
Manhattan plot of a genome-wide association study based on linear model for fiber diameter at second shearing of all the alpacas. The y-axis represents the negative base 10 logarithm of *p*-value. The x-axis represents the relative position of each SNP across the genome considering the order of scaffolds and chromosomes on VicPac3.1 assembly report.

**Figure 2 animals-13-03316-f002:**
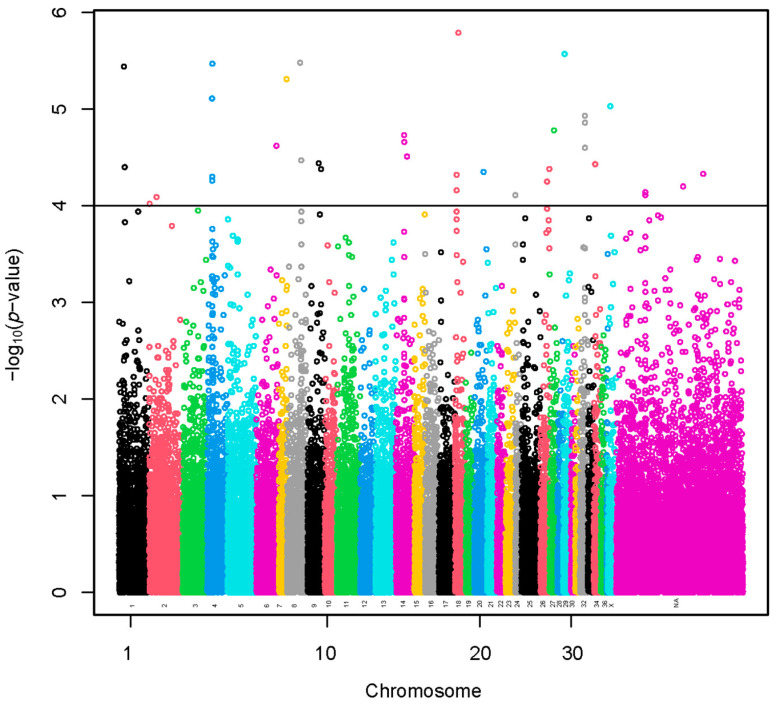
Manhattan plot of a genome-wide association study based on linear model for fiber diameter at second shearing of extreme groups. The y-axis represents the negative base 10 logarithm of *p*-value. The x-axis represents the relative position of each SNP across the genome considering the order of scaffolds and chromosomes on VicPac3.1 assembly report.

**Figure 3 animals-13-03316-f003:**
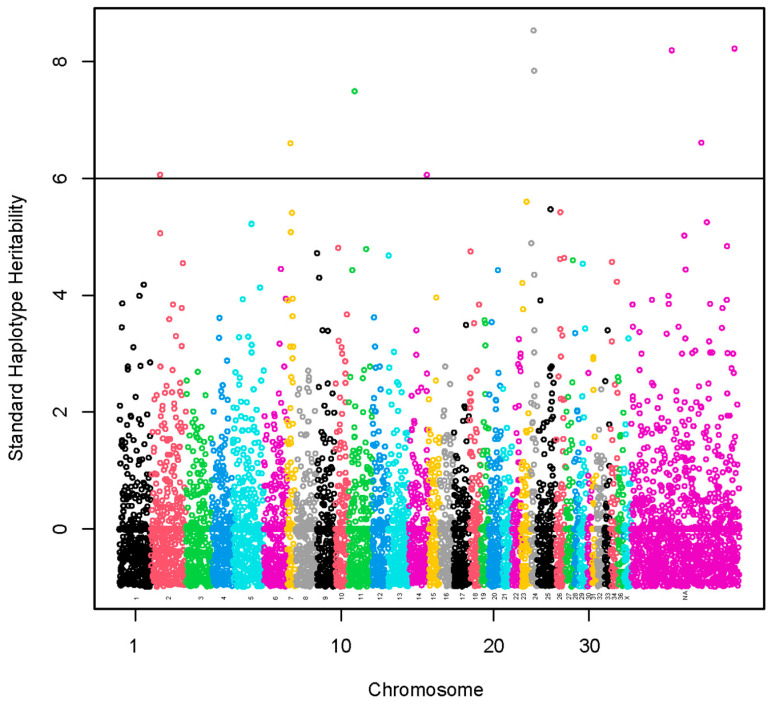
Manhattan plot of a genome-wide association study using haplotype and marker analysis for fiber diameter at second shearing. The y-axis represents haplotype heritability in units of standard deviations. The x-axis represents the relative position of each SNP across the genome considering the order of scaffolds and chromosomes on VicPac3.1 assembly report.

**Figure 4 animals-13-03316-f004:**
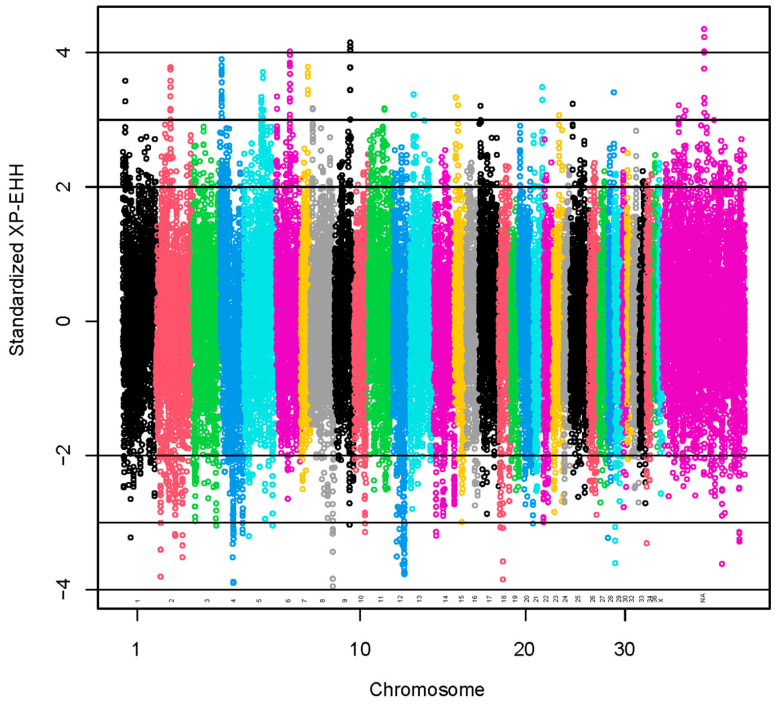
Manhattan plot for results of selection signatures for fiber diameter at second shearing. The y-axis represents standardized XP-EHH. The x-axis represents the relative position of each SNP across the genome considering the order of scaffolds and chromosomes on VicPac3.1 assembly report.

**Figure 5 animals-13-03316-f005:**
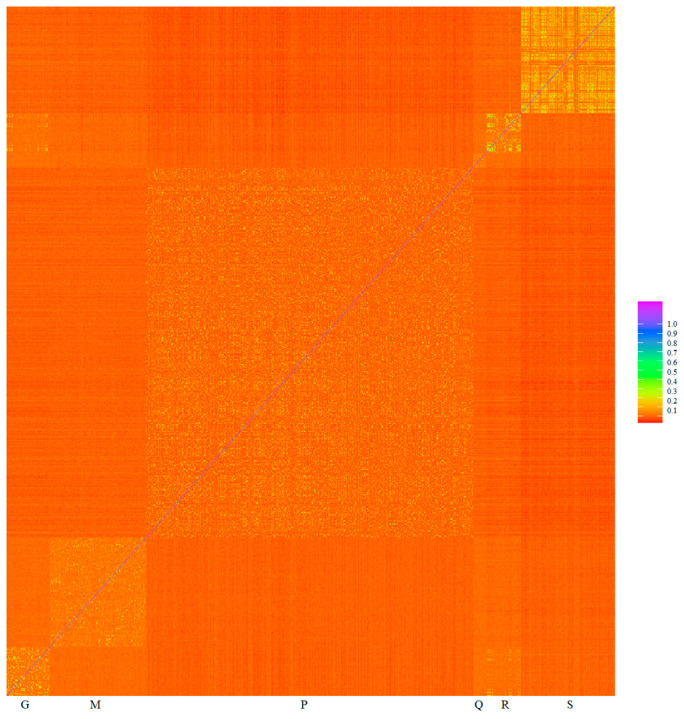
Heat map of genomic relationships among animals based on microarray genotyping of Gacocen (G), Mallkini (M), Pacomarca (P), Quimsachata (Q), Racco (R) and Sogamu (S).

**Table 1 animals-13-03316-t001:** Regions flanked by SNPs identified using two methods.

VPA	Scaffolds	Start	End	Methods	SNPs
6	ABRR03077257.1	41319229	41714500	Linear model—All the alpacas and selection signatures	AX-468770712 AX-417339934 AX-468704215 (*) AX-468766588 AX-417339050 AX-468763899 AX-468788057 AX-468767891 AX-417339948 AX-417339057 AX-468770136 AX-417339951 AX-468709166
9	ABRR03004037.1	19580657	19719247	Linear model—Extreme alpacas and selection signatures	AX-417305708 AX-468712525
29	ABRR03000003.1	4613849	4659732	Linear model—Extreme alpacas and selection signatures	AX-468751446 AX-417267477 AX-417266118 (*)
na	ABRR03000033.1	13561792	14451734	Linear model—All the alpacas and extreme alpacas	AX-417286284 AX-432730912 AX-417286293 AX-417284656 AX-417280615 AX-417286315 AX-417284681

* Common SNPs between both methods.

**Table 2 animals-13-03316-t002:** SNPs identified in alpacas are close to significant fiber-associated orthologous genes reported in goats.

VPA	Gene Name	Alpaca Scaffold	SNP	Distance (bp) Gene Marker	Method
2	*TACC3*	ABRR03000026.1	AX-417277871	289,181	Linear model—Extreme alpacas
7	*PSMA2*	ABRR03008474.1	AX-468769383	279,078	Linear model—Extreme alpacas
7	*POLD2*	ABRR03008474.1	AX-468769383	446,076	Linear model—Extreme alpacas
27	*NTRK3*	ABRR03000011.1	AX-468719480	127,219	Linear model—Extreme alpacas

## Data Availability

The data presented in this study are available at https://www.ebi.ac.uk/biostudies/studies/S-BSST1186.

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
