# Peer review of "Genome-Wide Association Study of Fiber Diameter in Alpacas"

_animals, 2023, doi:10.3390/ani13213316_

Round 1
Reviewer 1 Report
This manuscript is overall well-written and supports the use of the newly developed alpaca SNP chip in studies aimed at identifying the genetic basis of economically and biologically important traits in alpacas. I have a couple of broad questions before I described more detailed comments below:
- What was the purpose of the linear model GWAS including all the alpacas, and how do the authors believe putatively significant SNPs would explain an association with fiber diameter? The fiber diameter phenotypes of these individuals were not clearly described, and without a case/control assignment to the population, I believe that the validity of performing a GWAS can be questioned. Unless I misunderstood the authors’ purpose on performing this experiment.
- Nothing was said about how the fiber diameter phenotypes were obtained, only that “they were available”. Without many details about these phenotypes, which constitute arguably the most important piece of the puzzle, it is a bit difficult to interpret the results. For example, what was the range of diameters observed? Was diameter correlated with fiber color, or sex? How was diameter measured? Etc.
Some minor additional comments:
Line 22: “and three alpaca farms within region” – which region?
Line 28: I would suggest replacing “ensemble” with “designed” or maybe “developed”.
Line 31: “and three alpaca farms within region” – which region?
Lines 34-36: “The GWAS analysis based on a linear model identified 11 and 35 not significant 34 SNPs with p-value less than 1 x 10-4 using all the alpacas and alpacas with extreme values of fiber 35 diameter.” – What is the purpose of this sentence?
Line 55: I would suggest adding that this is the alpaca reference genome – “The alpaca reference genome has been assembled into 77,389 scaffolds (VicPac3.1, 55 GCA_000164845.4, NCBI)”.
Line 55: The reference cited lists a ∼190X genome coverage of short-read Illumina data.
Line 55: The reference cited lists 77,390 scaffolds (Table 1).
Lines 67-68: Would it be “the discovery of regulatory and/or structural variants in genes involved in hair follicle development”?
Lines 84-85: I would suggest adding that these are goat breeds.
Lines 89-90: Would it be “…the limited number of available SNP genotyping platforms”?
Line 162: “however, not significant SNPs with the lowest p-values were selected”. Can you please clarify this statement?
Line 188: I would suggest using “The genes within these haplotypes”
Line 215: Were these adjacent significant SNPs?
Line 239: I believe you mean to say that these 11 SNPs were significant and their –log10(p-value) was greater than 4, according to Figure 1 and Table S1.
Line 240: What do you mean by “the reported negative effects”? Please clarify.
Line 246 (Figure 1): Please provide the X-axis title.
Lines 251-252: Please reference Table S1.
Line 252-253: Please reference Table S2.
Line 254: Again, are these supposed to be significant SNPs? I believe there is confusion when referring to significant SNPs (p-value less than 1 x 10-4) in this manuscript.
Lines 255-257: Please clarify this sentence.
Line 262 (Figure 2): Please provide the X-axis title.
Line 291: were assembled? How?
Line 302 (Figure 3): Please provide the X-axis title.
Section 4.5. Candidate regions (lines 491-518: I believe the authors should be careful about making inferences on chromosome synteny between goats and alpacas without performing Zoo-FISH experiments to validate the synteny blocks reported in this section. Did the authors BLAST/BLAT the alpaca reference genome to confirm the chromosomal assignments?
Nothing major, just some minor grammar and spelling mistakes that can be fixed. For example, 'ensemble' instead of 'assembled' throughout the manuscript, among others.
Author Response
Comments 1
- What was the purpose of the linear model GWAS including all the alpacas, and how do the authors believe putatively significant SNPs would explain an association with fiber diameter? The fiber diameter phenotypes of these individuals were not clearly described, and without a case/control assignment to the population, I believe that the validity of performing a GWAS can be questioned. Unless I misunderstood the authors’ purpose on performing this experiment.
Response 1
The aim of the analysis was to determine genetic markers and genomic regions with a significant effect on the fiber diameter. We performed a linear model without a case/control assignment because fiber diameter is a quantitative trait. Moreover, fiber diameter is probably a polygenetic trait influenced by multiple genes. The use of a case/control analysis is frequently used for diseases or traits influenced by a small number of loci. Although the population could be classified as fine or coarse, as performed in method 3, however, we consider that the classification threshold is arbitrary. Likewise, after the formation of case and control groups, the number of animals retained in the analysis is significantly reduced, which removes genetic markers, specially those with low allelic frequency.
We have modified the text to specify the information of fiber diameter.
Line 106: “All the alpacas had white coat phenotype.”
Line 107-109: “The fiber diameter was evaluated using the Sirolan Laserscan according to International Wool Textile Organization standards (IWTO-12).”
Line 240-241: “The fiber diameter at second shearing of 960 samples was 20.53 ± 2.17 microns, and ranged from 13.73 to 29.50 microns.”
Comments 2
- Nothing was said about how the fiber diameter phenotypes were obtained, only that “they were available”. Without many details about these phenotypes, which constitute arguably the most important piece of the puzzle, it is a bit difficult to interpret the results. For example, what was the range of diameters observed? Was diameter correlated with fiber color, or sex? How was diameter measured? Etc.
Response 2
We have modified the text to specify the information of fiber diameter.
Line 106: “All the alpacas had white coat phenotype.”
Line 107-109: “The fiber diameter was evaluated using the Sirolan Laserscan according to International Wool Textile Organization standards (IWTO-12).”
Line 240-241: “The fiber diameter at second shearing of 960 samples was 20.53 ± 2.17 microns, and ranged from 13.73 to 29.50 microns.”
We did not take in account the correlation of fiber diameter with fiber color or sex because the sample includes only female Huacaya alpacas, and all the alpacas had white coat phenotype.
Some minor additional comments:
Comments 3
Line 22: “and three alpaca farms within region” – which region?
Response 3
We have modified the text to specify the regions.
Line 22-23: “two geographical Andean regions in Peru (Pasco and Puno), and three alpaca farms within each region.”
Comments 4
Line 28: I would suggest replacing “ensemble” with “designed” or maybe “developed”.
Response 4
We have replaced “ensemble” with “developed”.
Line 28: “using a recently developed alpaca SNP microarray”
Comments 5
Line 31: “and three alpaca farms within region” – which region?
Response 5
We have modified the text to specify the regions.
Line 31-32: “two geographical Andean regions in Peru (Pasco and Puno), and three alpaca farms within each region.”
Comments 6
Lines 34-36: “The GWAS analysis based on a linear model identified 11 and 35 not significant 34 SNPs with p-value less than 1 x 10-4 using all the alpacas and alpacas with extreme values of fiber 35 diameter.” – What is the purpose of this sentence?
Response 6
None significant SNPs were identified using a linear model, because their -log10(p-values) were lower than the significance threshold (6.01). However, some SNPs were selected as candidate markers by lowering the significance threshold to -log10(p-values) > 4 [49].
We have modified the text for better understanding.
Line 34-36: “The GWAS based on a linear model allowed to identified 11 and 35 SNPs (-log10(p-values) > 4) using information of all alpacas and alpacas with extreme values of fiber diameter, respectively.”
Comments 7
Line 55: I would suggest adding that this is the alpaca reference genome – “The alpaca reference genome has been assembled into 77,389 scaffolds (VicPac3.1, 55 GCA_000164845.4, NCBI)”.
Response 7
We have added “reference” in the text.
Line 55-56: “The alpaca reference genome has been assembled into 77,390 scaffolds (VicPac3.1, GCA_000164845.4, NCBI)”
Comments 8
Line 55: The reference cited lists a ∼190X genome coverage of short-read Illumina data.
Response 8
We have modified the genome coverage according to the cited reference.
Line 54-55: “Two research groups have sequenced the alpaca genome generating depths of 72.5X [1] and ∼190X [2]”
Comments 9
Line 55: The reference cited lists 77,390 scaffolds (Table 1).
Response 9
We have modified number of scaffolds according to the cited reference.
Line 55-56: “The alpaca reference genome has been assembled into 77,390 scaffolds (VicPac3.1, GCA_000164845.4, NCBI)”
Comments 10
Lines 67-68: Would it be “the discovery of regulatory and/or structural variants in genes involved in hair follicle development”?
Response 10
We have added “variants in” in the text.
Line 68-69: “the discovery of regulatory and/or structural variants in genes involved in hair follicle development”
Comments 11
Lines 84-85: I would suggest adding that these are goat breeds.
Response 11
We have added “goat breeds” in the text.
Line 85-86: “a non-cashmere goat breed and two cashmere goat breeds”
Comments 12
Lines 89-90: Would it be “…the limited number of available SNP genotyping platforms”?
Response 12
We have added “genotyping platforms” in the text.
Line 91: “the limited number of available SNP genotyping platforms”
Comments 13
Line 162: “however, not significant SNPs with the lowest p-values were selected”. Can you please clarify this statement?
Response 13
We have modified the text for better understanding.
Line 163-165: “The SNPs with -log10(p-values) lower than significance threshold (6.01) were considered not significant SNPs. Moreover, the SNPs (-log10(p-values) > 4) were selected as candidate SNPs”
Comments 14
Line 188: I would suggest using “The genes within these haplotypes”
Response 14
We have modified the text.
Line 191: “The genes within these haplotypes”
Comments 15
Line 215: Were these adjacent significant SNPs?
Response 15
The adjacent SNPs were not statistically significant. None significant SNPs were identified using a linear model, because their -log10(p-values) were lower than the significance threshold (6.01). However, some SNPs were selected as candidate markers by lowering the significance threshold to -log10(p-values) > 4 [49]. On Method 2, the haplotypes with the highest standardized heritability (> 6) were selected. On Method 3, the selection signatures with standardized values of XP-EHH > | 3 | were selected. These SNPs were used for the identification of candidate regions.
Comments 16
Line 239: I believe you mean to say that these 11 SNPs were significant and their –log10(p-value) was greater than 4, according to Figure 1 and Table S1.
Response 16
We have modified the text for better understanding.
Line 244-245: “A set of 11 SNPs (-log10(p-values) > 4) were selected using a sample of 960 alpacas (Figure 1 and Table S1).”
Comments 17
Line 240: What do you mean by “the reported negative effects”? Please clarify.
Response 17
We have modified the text for better understanding.
Line 245-247: “The SNP effects with values preceded by a negative sign (Table S1) indicate that the presence of the less frequent allele could be associated with a decrease in fiber diameter.”
Comments 18
Line 246 (Figure 1): Please provide the X-axis title.
Response 18
We have added the X-axis title.
Line 251 (Figure 1): “Chromosome”
Comments 19
Lines 251-252: Please reference Table S1.
Response 19
We have added Table S1 as reference.
Line 256-257: “A region of 290.9kbp with five identified SNPs was located at a chromosomally unassigned scaffold (Table S1)”
Comments 20
Line 252-253: Please reference Table S2.
Response 20
We have added Table S2 as reference.
Line 257-258: “A total of 3 lncRNA and 5 protein-coding genes harbored these SNPs or were found within a span of 500kbp from the identified SNPs (Table S2).”
Comments 21
Line 254: Again, are these supposed to be significant SNPs? I believe there is confusion when referring to significant SNPs (p-value less than 1 x 10-4) in this manuscript.
Response 21
We have modified the text for better understanding.
Line 259-260: “A set of 35 SNPs (-log10(p-values) > 4) were selected using the sample representing extreme groups of fiber diameter (Figure 2 and Table S1).”
Comments 22
Lines 255-257: Please clarify this sentence.
Response 22
We have modified the text for better understanding.
Line 260-262: “The SNP effects with values preceded by a negative sign (Table S1) indicate that the presence of the less frequent allele could be associated with a decrease in fiber diameter.”
Comments 23
Line 262 (Figure 2): Please provide the X-axis title.
Response 23
We have added the X-axis title.
Line 268 (Figure 2): “Chromosome”
Comments 24
Line 291: were assembled? How?
Response 24
We have modified the text for better understanding.
Line 296-298: “Hence, a total of 6,244 haplotype blocks (Table S5) including 20,438 SNPs located on 193 scaffolds were defined as previously described in section “Method 2. Haplotype and Marker Analysis”.”
Comments 25
Line 302 (Figure 3): Please provide the X-axis title.
Response 25
We have added the X-axis title.
Line 309 (Figure 3): “Chromosome”
Comments 26
Section 4.5. Candidate regions (lines 491-518: I believe the authors should be careful about making inferences on chromosome synteny between goats and alpacas without performing Zoo-FISH experiments to validate the synteny blocks reported in this section. Did the authors BLAST/BLAT the alpaca reference genome to confirm the chromosomal assignments?
Response 26
We have modified the text.
Line 520-521: "However, we must emphasize that the development of Zoo-FISH experiments is necessary to validate these synteny blocks."
We didn't BLAST/BLAT the alpaca reference genome to confirm the chromosomal assignments.
Reviewer 2 Report
The authors first genotyped alpacas’ DNA samples on a commercial Alpaca genotyping array (76K SNPs), applied three association study methodologies to the data, and finally identified four candidate regions with adjacent SNPs reported by two methods. As the first GWAS analysis of fiber diameter in alpaca using microarray data, this work is novel and interesting to the community, however, the overall quality of the manuscript could be further improved.
Major:
1. Figures 1-4 need labels for different colors, i.e., which color represents which genomic region (VPA/chromosome), as the x-axis of a typical Manhattan plot.
2. What is the full name of VPA? Instead of chromosome number as annotated by VicPac 3.1 for alpaca (Vicugna pacos) which have 37 chromosomes for Alpacas genome, why do the authors use VPA instead for chromosome annotation, but still use VicPac3.1 for SNP annotation?
3. As the first GWAS analysis using alpaca microarray data, authors need to provide guidance for the next steps, i.e., what experiments/analysis need to validate or further investigate the candidate regions, with the purpose of building the genomic tool for improving alpaca fiber quality.
4. Authors could consider a Venn diagram showing the SNPs identified by each method and how many SNPs were found in two or three methods.
Minor:
1. Can authors mark the threshold line in each figure?
2. Lines 538-539, typo: “VPA7 and VPA27 were localized within o close to…”
3. Lines 541-542, “The alpaca SNP array successfully identified alpaca SNPs and chromosomal regions associated to fiber diameter. “ should be "associated with."
Please see my comments above.
Author Response
Major:
Comments 1
- Figures 1-4 need labels for different colors, i.e., which color represents which genomic region (VPA/chromosome), as the x-axis of a typical Manhattan plot.
Response 1
We have added the labels for different colors and the X-axis in Figures 1-4.
Line 251 (Figure 1)
Line 268 (Figure 2)
Line 309 (Figure 3)
Line 323 (Figure 4)
Comments 2
- What is the full name of VPA? Instead of chromosome number as annotated by VicPac 3.1 for alpaca (Vicugna pacos) which have 37 chromosomes for Alpacas genome, why do the authors use VPA instead for chromosome annotation, but still use VicPac3.1 for SNP annotation?
Response 2
VPA is the nomenclature for chromosomes of Vicugna pacos as reported by Mendoza et al. [6]. We used VPA and the chromosome number in the manuscript.
Comments 3
- As the first GWAS analysis using alpaca microarray data, authors need to provide guidance for the next steps, i.e., what experiments/analysis need to validate or further investigate the candidate regions, with the purpose of building the genomic tool for improving alpaca fiber quality.
Response 3
We have added the Section "4.7. Future studies" in the manuscript.
Line 540-548:
“4.7. Future studies
The validation of the identified genetic markers should be the next step. Subsequent GWAS should incorporate a higher number of alpacas to confirm the association of SNPs with fiber diameter. Moreover, since the alpaca herds were poorly genetic connected [15] both the allelic frequency of SNPs and the association with fiber diameter in other populations must be evaluated, in order to know whether the markers are valid for any population or only for the populations studied. After the validation of the genetic markers, genotyping tests must be developed to applied genomic selection for reducing fiber diameter.”
Comments 4
- Authors could consider a Venn diagram showing the SNPs identified by each method and how many SNPs were found in two or three methods.
Response 4
The number of SNPs by each method and how many SNPs were common between methods is included in Table S12.
Minor:
Comments 5
- Can authors mark the threshold line in each figure?
Response 5
The figures 1-4 has the threshold line.
Line 251 (Figure 1)
Line 268 (Figure 2)
Line 309 (Figure 3)
Line 323 (Figure 4)
Comments 6
- Lines 538-539, typo: “VPA7 and VPA27 were localized within o close to…”
Response 6
We have modified the text.
Line 554-556: “SNPs located on VPA2, VPA7 and VPA27 and identified by one association method of analysis were localized within or next to orthologous alpaca genes”
Comments 7
- Lines 541-542, “The alpaca SNP array successfully identified alpaca SNPs and chromosomal regions associated to fiber diameter. “ should be "associated with."
Response 7
We have replaced “associated to” with “associated with”.
Line 557-558: “The alpaca SNP array successfully identified alpaca SNPs and chromosomal regions associated with fiber diameter”
Round 2
Reviewer 1 Report
Thank you for addressing my comments. I feel satisfied with the answers provided and with the corrections to the manuscript.
Reviewer 2 Report
Thank the authors for addressing my questions.